# Ciguatera-Causing Dinoflagellate *Gambierdiscus* spp. (Dinophyceae) in a Subtropical Region of North Atlantic Ocean (Canary Islands): Morphological Characterization and Biogeography

**DOI:** 10.3390/toxins11070423

**Published:** 2019-07-19

**Authors:** Isabel Bravo, Francisco Rodriguez, Isabel Ramilo, Pilar Rial, Santiago Fraga

**Affiliations:** Centro Oceanográfico de Vigo, Instituto Español de Oceanografía (IEO), Subida a Radio Faro 50, 36390 Vigo, Spain

**Keywords:** *Gambierdiscus*, benthic dinoflagellates, Canary Island, CFP, ciguatera

## Abstract

Dinoflagellates belonging to the genus *Gambierdiscus* produce ciguatoxins (CTXs), which are metabolized in fish to more toxic forms and subsequently cause ciguatera fish poisoning (CFP) in humans. Five species of *Gambierdiscus* have been described from the Canary Islands, where CTXs in fish have been reported since 2004. Here we present new data on the distribution of *Gambierdiscus* species in the Canary archipelago and specifically from two islands, La Palma and La Gomera, where the genus had not been previously reported. *Gambierdiscus* spp. concentrations were low, with maxima of 88 and 29 cells·g^−1^ wet weight in samples from La Gomera and La Palma, respectively. Molecular analysis (LSUrRNA gene sequences) revealed differences in the species distribution between the two islands: only *G. excentricus* was detected at La Palma whereas four species, *G. australes*, *G. caribaeus*, *G. carolinianus*, and *G. excentricus*, were identified from La Gomera. Morphometric analyses of cultured cells of the five Canary Islands species and of field specimens from La Gomera included cell size and a characterization of three thecal arrangement traits: (1) the shape of the 2′ plate, (2) the position of Po in the anterior suture of the 2′ plate, and (3) the length–width relationship of the 2″″ plate. Despite the wide morphological variability within the culture and field samples, the use of two or more variables allowed the discrimination of two species in the La Gomera samples: *G.* cf. *excentricus* and *G.* cf. *silvae*. A comparison of the molecular data with the morphologically based classification demonstrated important coincidences, such as the dominance of *G. excentricus*, but also differences in the species composition of *Gambierdiscus*, as *G. caribaeus* was detected in the study area only by using molecular methods.

## 1. Introduction

*Gambierdiscus* is a genus of benthic dinoflagellates that produces ciguatoxins (CTXs) and maitotoxins (MTXs). The transfer of CTXs up the food chain results in their metabolism and accumulation in the tissues of fish, including edible species, thus potentially causing ciguatera fish poisoning (CFP) in humans. However, the dynamics of the trophic transfer of CTXs remains unclear. Biotransformations that occur in fish have been reported to play a role in the accumulation and retention of toxins identified in *G. polynesiensis* from the Pacific Ocean [1,2]. However, with the exception of this species, there is no information on the profile of CTXs in *Gambierdiscus*. Investigations of the occurrence of CFP requires accurate knowledge of the species present in areas where the disease appears. Thus, determinations of the biogeography of this genus must be complemented by a detailed morphological comparison between *Gambierdiscus* species.

The presence of CTX-producing dinoflagellates in tropical and subtropical regions, such as the Caribbean Sea, Hawaiian Islands, Southeast Asia, French Polynesia, Indian Ocean, and tropical and subtropical Australia, is well documented [3,4,5]. Over the last decade, however, the incidence of CFP in some of those areas has been increasing, as reported for the Pacific Islands [6]. Moreover, new observations of *Gambierdiscus* in subtropical-temperate regions such as the Canary Islands, where CFP has been reported since 2004, is an emerging problem [7,8]. In addition, *Gambierdiscus* species have recently been detected in areas characterized by a temperate climate, including the Mediterranean Sea, Gulf of Mexico, temperate areas of Japan, Brazil, and the coast of North Carolina [9,10,11,12,13]. The reasons for this apparent expansion are still under discussion. Although it no doubt in part reflects a more geographically intense sampling during the last few years. Climate change has been implicated as well [14], and an association between CFP incidence and a warm sea surface temperature has been proposed [15]. Consequently, global warming can be expected to cause important changes in the diversity, abundance and distribution of *Gambierdiscus* species during this century.

Among the 16 species of *Gambierdiscus* described so far, five have been detected in the Canary Islands: *G. australes*, *G. caribaeus*, *G. carolinianus*, *G. excentricus*, and *G. silvae* [12]; the two last species were originally described from those islands [16,17]. Although the first recorded case of ciguatera in the region was relatively recent, in 2004 [8,18], and *Gambierdiscus* spp. (subsequently identified as *G. excentricus* by [17]) was also detected for the first time in Tenerife in 2004 [9], both the diversity and the concentrations of *Gambierdiscus* spp. (average of up to 2200 cells·g^−1^ in one location) within the archipelago resulted to be quite high, as noted by [12]. The increase in CFP outbreaks, positive CTXs detection in fish, and the presence of a high diversity and concentration of *Gambierdiscus* have generated considerable concern regarding the regional persistence and spread of CFP [8,12]. Although more specific studies are needed, Caribbean CTX (C-CTX1) has been reported to be mainly responsible for CFP contamination in fish samples from the East Atlantic coast related to ciguatera [7,19,20,21].

Progress in studies of the ecology and biogeography of *Gambierdiscus* has been hindered by the difficulty in morphologically differentiating among species, especially using traditional light or electron microscopy methods [22,23]. Rather, the unequivocal identification of *Gambierdiscus* cells requires genetic sequencing techniques, most of which are currently based on the ribosomal DNA sequences of cultured cells. Semi-quantitative molecular techniques have been used for five species of *Gambierdiscus* [24], but quantitative determinations of specimens in field samples remains challenging. However, because *Gambierdiscus* species differ in their toxicity [25,26], an assessment of the overall risk of CFP rests on the ability to differentiate among them. In fact, *G. excentricus*, first described in Canary Islands in 2004 and the most abundant species reported in the region so far, has been recently shown as one of the most toxic species known to date, both for CTXs and MTXs [25,26,27]. Thus, an objective of the present study was to provide a morphological comparison of the species that have been detected in the Canary Islands and therefore evaluate the extent to which a morphologically based taxonomy can be used in the identification of *Gambierdiscus* species in the waters of the region.

The morphological identification of thecate dinoflagellates is commonly based on cell size and shape together with the architecture of the thecal plates. However, the morphological differences among *Gambierdiscus* species are often very subtle [23,28]. A detailed morphological comparison of nine species was provided by [23]. Three of those species were included in the present study, *G. australes*, *G. caribaeus*, and *G. carolinianus*, together with two others also identified in the Canary Islands. Both cultures and field specimens were used in our analysis. In addition, for a more complete characterization of the biogeography of *Gambierdiscus* in the Canary archipelago, the diversity and abundance of this genus in La Gomera and La Palma were determined. These are the only two Canary Islands where this genus has not yet been reported—with the exception of *G. excentricus* in La Palma, detected during opportunistic sampling (Fraga, unpublished observations). 

## 2. Results

### 2.1. Diversity and Abundance of Gambierdiscus from La Palma and La Gomera

LSUrRNA gene sequencing of 61 isolates, 3 from La Gomera and 58 from La Palma, showed that *G. excentricus* was the only species identified from La Palma, whereas four species, *G. australes*, *G. caribaeus*, *G. carolinianus*, and *G. excentricus*, were identified from La Gomera, albeit in very different proportions (Figure 1). Thus, of the 40 *Gambierdiscus* strains isolated from macrophytes from this last island, 65% were *G. excentricus*, followed by *G. silvae* (18%), *G. caribaeus* (15%), and *G. australes* (2%) (Figure 1c). In addition, *G. caribaeus* (18 isolates) was detected in a net sample collected in the harbor of Playa Santiago, in La Gomera. No *Gambierdiscus* cells were detected in any of the samples taken from other harbors (Table 1).

The average concentration of *Gambierdiscus* spp. in macrophytes was 2 cells·g^−1^ in samples from La Palma (SD = 5, n = 25) and 12 cells·g^−1^ in those from La Gomera (SD = 21, n = 28). The maximum concentrations were also observed in the latter with 88 cells·g^−1^ in La Cueva beach (La Gomera) by contrast with 29 cells·g^−1^ in Tazacorte beach (La Palma) (Table 1, Figure 1). The average abundance at each station is shown in Figure 1b. *Gambierdiscus* cells appeared in 17 of the 28 La Gomera samples but in only 3 of the 25 La Palma samples. A comparison of macrophyte weights measured as the wet weight vs. the dry-blot weight indicated a wet-weight loss of 62% (SD = 14, n = 53), which translated into an increase in the concentrations when expressed as the blot-dried weight. Accordingly, the maximum *Gambierdiscus* concentration of 88 cells·g^−1^ wet weight corresponds to 130–155 cells·g^−1^ blot-dried weight of macrophytes.

### 2.2. Epiphytic Dinoflagellate Composition

The concentrations of dinoflagellates of other genera besides *Gambierdiscus* were also quantified in the epiphytic samples from La Gomera and La Palma and included *Prorocentrum, Coolia, Sinophysis, Ostreopsis, Vulcanodinium, Heterocapsa*, and *Scrippsiella* (Appendix A). The most abundant dinoflagellates were those of the *Ostreopsis* genus, which reached concentrations of 2.8 × 10^3^ and 1.6 × 10^3^ cells·g^−1^ in samples from Tazacorte beach (La Palma) and Charco Condesa (La Gomera), respectively. While the concentrations of *Gambierdiscus* were low, the cells were detected in a higher number of samples than either *Heterocapsa* or *Scripsiella* (Appendix A). The principal component analysis (PCA) showed significant differences between islands with respect to the composition and abundances of the studied dinoflagellates (*p* < 0.001), with the two principal components explaining 46% of the variance. The rotated component matrix is shown in Table 2 (absolute values <0.5 were excluded). In the first component, the highest correlation was between *Gambierdiscus* and *Sinophysis*, and secondly between *Prorocentrum, Coolia*, and *Vulcanodinium*. Based on the correlation coefficients, the first component was significantly more important in La Gomera than in La Palma. The second component assembled the remaining genera, *Ostreopsis*, *Heterocapsa*, and *Scrippsiella* (Table 2), but without a clear correlation with any of the islands.

### 2.3. Morphological Study of Cultured Gambierdiscus Cells

Cells of cultures of the five *Gambierdiscus* species from the Canary Islands were anteroposteriorly compressed, and their thecal plate formula was typical of the genus: Po, 4′, 0a, 6″, 6c, 5‴, 0p, 2″″. The cell sizes and thecal plate measurements of the species were compared. The very small cells of *G. silvae* allowed this species to be differentiated based on a definition of >50% of the cells being ≤60 µm in diameter (Table 3, Figure 2a,b). The cell size distributions of the other species also differed significantly (*p* < 0.01); the exceptions were *G. caribaeus* and *G. carolinianus*, as indicated by their very similar mean ranks (Appendix A). Observations of the shape of the 2′ and 2″″ plates based on R1, R2, and R3 (see Materials and Methods) also revealed significant differences among species (*p* < 0.01). Only the comparison of the R3 distribution according to the mean rank showed no significant differences between *G. caribaeus* and *G. silvae* (*p* > 0.01) (Appendix A). The values of the three parameters are box-plotted in Figure 2c–e. 

*G. australes* cells had a mean (± SD) depth of 81 ± 6.3 µm and a mean width of 78 ± 7.5 µm. Among the studied species, the 2″″ plate of *G. australes* was the most elongated (Figure 2e and Figure 3a), resulting in the highest R3 value (R3 > 1.5, mean ± SD = 1.91 ± 0.20) (Table 3). For the 2′ plate of *G. australes*, the R1 value was close to 1 (mean ± SD = 0.72 ± 0.13, Table 3) and second highest after the R1 of *G. caribaeus* (mean ± SD = 0.83 ± 0.11, Table 3), which indicated the more rectangular shape of this plate in these two species (Figure 3b,c) than in the others, in which the smaller R1 values reflected a more hatchet-shaped 2′ plate. The R2 values of *G. australes* (mean ± SD = 1.67 ± 0.16) differed significantly from those of *G. excentricus*, with only a 5% overlap (Table 3 and Figure 2d). 

*G. caribaeus* cells had a mean (± SD) depth and width of 76 ± 7.2 µm and 74 ± 6.7 µm, respectively. Other distinctive characteristics of this species, in addition to its more rectangular 2′ plate, as described above, was its R3 value (mean ± SD = 1.44 ± 0.14) indicating less elongation of the 2″″ plate compared to that of *G. australes* (Figure 3a,d). Lower R3 values distinguished not only *G. caribaeus* but also *G. carolinianus* and *G. silvae* from *G. australes* and to a lesser extent from *G. excentricus* (Table 3 and Figure 3a,d,f,h,j). Furthermore, the R2 values of *G. caribaeus* (mean ± SD = 1.34 ± 0.22, Table 3) significantly differed from those of *G. excentricus*, *G. australes*, and *G. silvae* but were very similar to the R2 values of *G. carolinianus*, as it is indicated by the similar mean ranks (Appendix A).

*G. carolinianus* cells had a mean (± SD) depth of 73 ± 4.6 µm and a mean width of 71 ± 4.9 µm. This species along with *G. silvae* had the smallest R1 mean values (mean ± SD = 0.50 ± 0.08) (Table 3 and Figure 2c), indicating a hatchet-shaped 2′ plate (Figure 3e,i) but different from the more rectangular of *G. australes* (Figure 3b) and *G. caribaeus* (Figure 3c). However, as stated above, the R2 values of *G. carolinianus* (mean ± SD = 1.44 ± 0.19) differed from those of *G. silvae* and *G. excentricus*, the species with the least and most eccentric Po, respectively (Table 3). The R3 values of *G. carolinianus* (mean ± SD = 1.31 ± 0.12) indicated less elongation of the 2″″ plate (Figure 3f), a characteristic shared, as noted above, with *G. caribaeus* and *G. silvae* (Table 3 and Figure 3d,j) but different from *G. australes* and *G. excentricus* (Figure 3a,h).

*G. excentricus* had the largest cells among the studied species (88 ± 6.8 µm depth and 83 ± 7.9 µm width, mean ± SD). A distinguishing feature of this species was its highly eccentric Po, evidenced by R2 ≥ 2.1 (mean ± SD = 2.55 ± 0.31, Table 3 and Figure 2d and Figure 3g). The sole species with an overlapping R2 was *G. australes*, but the overlap was only by 5%, as mentioned previously. A comparison of these two species showed that the hatchet-shaped 2′ plate frequently seen in *G. excentricus* (Figure 3g) (mean ± SD of R1 = 0.67 ± 0.11) contrasted with the mostly rectangular 2′ plate of *G. australes* (Figure 3b). In addition, the 2″″ plate was significantly more elongated in *G. australes* (Figure 3a) than in *G. excentricus* (mean ± SD of R3 = 1.71 ± 0.15, Table 3 and Figure 3h) though with a degree of overlapping values as shown in Figure 2e.

*G. silvae* was characterized by the smallest cells (60 ± 9.9 µm depth and 59 ± 9.7 µm width, mean ± SD) and an R2 value close to 1 (mean ± SD = 1.02 ± 0.16), indicating a nearly central position of Po in the anterior suture of the plate and representing a less eccentric Po (Table 3 and Figure 3i). This feature clearly distinguished *G. silvae* from *G excentricus* and *G. australes* (Figure 2d). The R3 values of *G. silvae* (mean ± SD = 1.41 ± 0.14, Table 3) were consistent with a less elongated 2″″ plate (Figure 3j), a feature shared with *G. caribaeus* and *G. carolinianus* but not with *G. australes* and *G. excentricus*, as has already been mentioned previously. 

### 2.4. Morphological Study of Field Cells of Gambierdiscus

*Gambierdiscus* specimens from field samples showed the same general morphology as the cultured cells (i.e., anteroposterior compression) and the same thecal plate formula. Field cells were classified based on cell depth (D) and comparisons of their R1, R2, and R3 values with those values determined for the cultured cells (Table 3). Thus, D and R2 separated two well-defined groups (Figure 4a), whereas the discriminatory powers of R1 and R3 were low (Figure 4b). Using D and R2, we were able to classify at least 38 out of 42 specimens from field samples into two groups, corresponding to *G.* cf. *excentricus* and *G.* cf. *silvae* (Figure 4a). The assignment of 28 specimens to the first group was based on D > 80 µm and R2 > 1.97. A cell with D > 80 µm but R2 = 1.9 (in Figure 4a it is marked with an arrow), such that it might have corresponded to either G. cf. *excentricus* or *G.* cf. *australes*, was assigned to the former because of the similarities in the size and shape of its 2′ plate. The assignment of eight specimens to *G.* cf. *silvae* was based on D < 61 µm and R2 < 1.41 (Figure 4a). In the classification of the other four cells, both R1 and/or R3 were additionally necessary as well as an evaluation of the general appearance of the cells. Three specimens had D and R2 values consistent with *G.* cf. *silvae*/*G.* cf. *carolinianus*/*G.* cf. *caribaeus* (64 ≤ D ≤ 66 and 1.21 ≤ R2 ≤ 1.31, Figure 4a), but their R1 and R3 values as well as the general appearance of the cells coincided with those of the *G.* cf. *silvae* group (Figure 4b). Lastly, one cell was ruled out as being either *G.* cf. *silvae* or *G.* cf. *excentricus* because of its D (81 µm) and R2 (1.49) values (Figure 4a); instead, its R1 (0.48) and R3 (1.27) values were consistent with those of *G.* cf. *carolinianus* (Figure 4b). 

### 2.5. Gambierdiscus Diversity: Genetics vs. Morphology

As shown by LSUrDNA sequencing and morphological study, *G. excentricus* was the most abundant species, detected in 70% of the sequences (23 of 33) (Figure 5a), and with an abundance of 72% (30 of 42 cells) as determined by morphology (Figure 5b). The corresponding values for *G. silvae* were 12% (4 of 33 sequences) (Figure 5a) and 26% (11 of 42 cells) (Figure 5b). *G. caribaeus*, by contrast, was detected only in the genetic study, in which it accounted for 18% (6 out of 33) of the sequences. One cell classified as *G.* cf. *carolinianus* (2%) was also identified as such in the morphological study.

## 3. Discussion

### 3.1. Geographical Distribution of the Species of Gambierdiscus Detected in the Archipelago of Canary Islands

Our results on the diversity of *Gambierdiscus* in La Gomera and La Palma complete the study of the diversity of this genus in the archipelago. The geographical distribution of the species recognized thus far is depicted in the map shown in Figure 6, which is the first presentation of the distribution of *Gambierdiscus* in the seven main islands of the Canary archipelago. The map is based on the results of [12] and those of the present study, and specifically on data of genetic sequences. These were mostly obtained from cultures established from cells isolated from field samples. However, the number of sequences corresponding to each island differed. 

The dominant species of *Gambierdiscus* in the Canary Islands were *G. excentricus* and *G. australes*. Unlike their fairly coincidental distribution of these species in the archipelago, literature data on their worldwide geographical distribution show marked differences. Thus, *G. australes* was originally described from coral reef areas near Raivavae island (French Polynesia) and was considered to be limited to the Pacific Ocean [22], whereas *G. excentricus* was quite recently described from macrophytes in rock pools of the Canary Islands [17], and it had not been observed in well-studied areas as in the Pacific and until very recently in Caribbean Sea [10,25]. Nonetheless, during the last decade, an increasingly larger distribution of these two *Gambierdiscus* species has been reported. For example, *G. australes* was reported from the Indian Ocean, along the coast of Pakistan (Arabian Sea) [29], and was recently detected off the coasts of the Balearic Islands (Mediterranean Sea) [30]. To what extent this expansion reflects intensified studies of benthic dinoflagellates is unclear, but it may explain the detection of *G. excentricus* following its original identification in Canary Island samples [17]. *G. excentricus* was also found in the Atlantic Ocean, both along the island of Madeira [31,32] and, relatively close by, in Morocco—note that the species corresponding to *Gambierdiscus* spp. cited by [33] corresponds to *G. excentricus* according to [17]. Other reported detections of *G. excentricus* include the coastal waters of Brazil (Playa Tartaruga, Rio de Janeiro) [34] and Salalah, Oman (Arabian Sea) (María Saburova, personal communication).

The distributions of *G. caribaeus*, *G. carolinianus*, and *G. silvae* in the Canary Islands are apparently more limited than those of the two aforementioned species [12]. Nonetheless, in our study *G. caribaeus* and *G. silvae* were quite commonly detected off La Gomera. Thus far, there has been only one report of *G*. *carolinianus* in Tenerife [12]. The same authors reported *G. caribaeus* on the island of El Hierro, where it was associated with macroalgae and an artificial substrate. A high concentration of *Gambierdiscus* cells (>10^4^ cells·g^−1^ wet weight of algae) was registered on the same island, at the port of La Restinga, and based on their morphology was identified as *G. caribaeus* [35]. We detected this species in the waters of La Gomera, associated with macroalgae and with sediment from an island harbor. Thus, compared to the other *Gambierdiscus* species, the distribution of *G. caribaeus* seems to be concentrated in the western islands. This may be due to a more tropical character of this species, given the warmer sea temperatures of La Gomera, as opposed to the cooler waters of Lanzarote and Fuerteventura. The worldwide distribution of *G. caribaeus* extends to throughout the Caribbean, including Florida (USA), but also to islands in the Pacific Ocean including Tahiti (French Polynesia) and Palau (Micronesia) [10]. In contrast, the worldwide distribution of *G. silvae* is thus far very restricted, as it is largely limited to the Atlantic Ocean. In addition to the Canary Islands, ribotype 1, from Belize [10], is coincident with *G. sylvae*, and it was also detected in the Cape Verde archipelago, as its morphological characteristics suggest that the species identified by [36] and cited as *Goniodoma* would be referred to today as *G. silvae* [16]. This species was also recently identified among *Gambierdiscus* specimens obtained in Greece (eastern Mediterranean) ([9]; Aligizaki K., personal communication).

### 3.2. Morphological Study 

Cell depth and the shapes of the second apical (2′) and second antapical (2″″) plates have been described by several authors as differential morphological features of *Gambierdiscus* species [16,17,22,23,37]. The relatively central position of Po was a differential trait noted by [17] in their original description of *G. excentricus*. Morphological characterization of ten *Gambierdiscus* species by [23], including G. *australes*, G. *carolinianus*, and G. *caribaeus*, provided a dichotomous key in which those three species were separated based on the shapes of their 2′ and 2″″ plates. The present work applied those morphological capabilities to differentiate among the five *Gambierdiscus* species isolated from the Canary Islands. Moreover, by applying the knowledge obtained from cultures to field specimens, we obtained a more detailed morphological characterization of these dinoflagellates in nature and an approach to cell determinations that will be of value, at least until new quantitative molecular techniques are developed.

The cell sizes of the studied species were highly variable, and although the distributions of cell depth (D) and width (W) differed significantly, the large degree of overlap between the respective size ranges hinders the use of either (D) or (W) in species identification. Of the five studied species, only the smaller size (≤ 60 µm) of *G. silvae* can be considered as a useful trait for species differentiation. In the morphological study of field specimens, it allowed—along with the morphology of the thecal plates—the identification of *G. silvae* in samples from Playa de Santiago. 

The high intraspecific variability of R1, R2, and R3, representing the shapes of 2′ and the 2″″ plates and the position of Po, demonstrated well the thecal shape variations within *Gambierdiscus*. This variability and the subtle differences between some species complicate a morphologically based discrimination. This problem has been repeatedly mentioned in the literature and explains why the first description of *Gambierdiscus* (*G. toxicus* [38]) included several similar species that could not be distinguished until molecular methodologies were used [22,23]. Morphological intraspecific variability has been attributed to the changes that occur during the division process, as even two recently divided daughter cells may differ in the cell shape [17]. In addition, the details observed in the present study suggest that morphological variability of the thecal plates occurs during cell division. For example, in recently divided cells of *G. australes*, the extent of 2″″plate elongation is far below the mean and is instead similar to that in *G. excentricus* (as it can be seen in the cell of *G. australes* showed in Figure 3k). In the cells of *G. excentricus*, the less dense upper end of the 2″″ plate suggests a progressive lengthening during cell growth, as indicated herein by R3 values well above the average for this species and contributing to the overlap with *G. australes* (as it can be seen in the cell of *G. excentricus* showed in Figure 3l). 

Among the 16 species of *Gambierdiscus* described so far, it is obvious that limiting the number of species to those that appear in a certain area would facilitate the comparison and identification of them. It must be always in mind that the appearance of a new species in the Canary Islands should require new comparative studies. We found that the variability in the morphological features of *Gambierdiscus* was also a characteristic of the field specimens, but the consideration of two or more traits and even the general morphological appearance of the cell resulted in a classification. For example, two cells of *G.* cf. *excentricus* had very high R3 values (>1.8), within the range determined for *G.* cf. *australes*. Nonetheless, one of the cells could be unequivocally identified based on the eccentricity of Po and the shape of the second apical plate. In the other cell, the eccentricity of Po (R2 = 1.90) was within the ranges of *G.* cf. *excentricus* and *G.* cf. *australes*, but the similarity of the general cell morphology allowed an assignment to the former species. 

Thus, despite the similarities among species from the Canary Islands, the parameters described and applied herein can be used in their discernment. While *G. excentricus* and *G. silvae* were easily discriminated among the five species, this may not be the case if other species are present in the samples. Accordingly, it will often be necessary to use more than one morphological trait to identify *Gambierdiscus* species. The parameters included in our study differed in their discriminatory power depending on the species. R2, representing the position of Po, had the best resolution, followed by R3 (elongation of the 2″″ plate). R1, indicating the shape of the 2′ plate, was also informative. Furthermore, we discovered that once the observer had become familiar with the traits of the different species, recognition of the general shape and configuration of the thecal plates could aid in the identification. In addition, the literature describes other potentially useful morphological characteristics of *Gambierdiscus*, for example, the symmetry of the 3″ plate, the dorsal end of the 2″″ plate, the cell surface pattern, and the shape of the apical pore complex (as examples see [11,16,23,34]). 

### 3.3. Comparisons of the Results: Genetics vs. Morphology 

The most important difference between the genetic and morphological analyses was the restriction of the detection of *G. caribaeus* to the genetic study, where it accounted for 18% of the obtained sequences. This difference demonstrates well the bias associated with each methodology. *G. caribaeus* occurred in just one sample (Playa Santiago), from which three other species were isolated (5 specimens of *G. caribaeus*, 2 of *G. excentricus*, and 1 of *G. silvae*). By contrast, all four specimens from the same sample that were studied morphologically could be assigned to *G.* cf. *silvae*. Consequently, either the cultures facilitated *G. caribaeus* detection or the morphological study was incorrect. This second possibility was ruled after we meticulously examined the morphological features of those four specimens and again classified them as *G. silvae* for the following reasons: (1) the small size of the specimens, (2) the hatchet-shaped 2′ plate, (3) the more centered position of Po, and (4) the short elongation of the 2″″ plate. Furthermore, the morphology of those specimens coincided with that of cells identified as *G.* cf. *silvae* by genetic analysis in other samples. 

Amplification of LSUrDNA fragments by routine PCR does not provide quantitative data on the actual species distribution, given that the results are based on cultured cells or single individuals. Consequently, PCR analyses will be biased by the fact that specimens or species better adapted to laboratory conditions will be overrepresented such that the selected single cells are unlikely to be representative of the overall population. In our results, the smaller size of *G.* cf. *silvae* might cause this species to be overlooked when it is present together with the larger *G. caribaeus* cells isolated for cultures. 

These observations point out the importance of semi-quantitative tools, as species-specific PCR assays [24], in estimating *Gambierdiscus* diversity. The results of those analyses in several geographical regions, such as the Gulf of Mexico and the Pacific [4,39,40], have revealed broad-ranging differences in species composition and in the relative proportions of the identified species. Similar studies in the Canary Islands will add to our knowledge of the spatio-temporal distribution and species composition of *Gambierdiscus* as well as to associate these trends with the prevailing environmental conditions and the detection of ciguatoxic fish. 

## 4. Materials and Methods

### 4.1. Field Sampling and Processing

Fifty-three samples of macrophytes were collected from the littoral of the islands La Gomera and La Palma in Canary archipelago at six and three stations, respectively, between 4–9 October 2017 (Table 1, Figure 1). Two to eight samples were collected from each station gathering a representative range of material from each sampling location. In addition, 11 phytoplankton samples were obtained by net tows from 4 harbors located in the close vicinity of those stations (Table 1). The macrophyte samples were collected during low tide from tidal ponds and by snorkeling up to a maximum depth of 1.5 m. The phytoplankton samples were filtered through a nylon mesh with a mesh size of 300 µm and then concentrated by filtration through a 20 µm nylon mesh. One set of subsamples was kept for cell isolations and culture establishment, and another was formaldehyde-fixed in situ for identification and enumeration in the laboratory. The methodology used in the cell isolations and culture establishment of *Gambierdiscus* spp. was that described in [12]. Among the 61 culture isolates used in our phylogenetic analysis of *Gambierdiscus*, 58 were from La Gomera and 3 from La Palma.

### 4.2. Cell Enumeration and Light Microscopy

Formaldehyde-fixed epiphyte samples were stained with Fluorescent Brightner 28 (Sigma, St Louis, MO, USA) [41] for thecal identification and counted under UV light using an Axiovert 125 epifluorescence inverted microscope (Carl Zeiss AG, Germany). Quantitative data were obtained for the genus *Gambierdiscus* and for the other dominant dinoflagellates also present in the samples. Cell abundances were expressed as cells per gram wet weight of the host macrophyte (abbreviated as cells·g^−1^). The macrophytes were weighed after they had been drained of their water by manual squeezing. This is the most commonly used measurement method, and it allowed comparisons of our results with those of other studies, although many do not provide a detailed description of the procedure. In addition, because previous studies reported data from the Canary Islands as cells per gram blot dry weight, we also weighed the sampled macrophytes after blotting them overnight on absorbent laboratory paper [12]. The correction factor between the two procedures was estimated as the difference in the average of the percentages calculated from the wet and blot-dried measured weights.

### 4.3. DNA Extraction, PCR Amplification, and DNA Sequencing

Partial LSUrRNA gene sequences (D8-D10 region) were obtained from cells isolated from cultures as described by [12]. Briefly, single cells were transferred using a micropipette to an Eppendorf 5424R tube (Eppendorf AG, New York, USA), washed in three droplets of distilled water, frozen in liquid nitrogen, and stored at −20 °C until processing (on the same day). The cells were rinsed in 1 mL of distilled water, centrifuged again, and the supernatant was discarded. DNA was extracted using a modified Chelex procedure as described in [16] and eluted in TE buffer (25 mL). The quantity and purity of the DNA were analyzed in a Nanodrop Lite spectrophotometer (Thermo Scientific, Waltham, MA, USA).

The D8–D10 regions of the LSUrRNA gene were amplified using the primer pair FD8/RB [42]. The amplification reaction (20 μL) contained 0.75 pmol of each primer/µL and 2 μL of DNA extract in Horse-Power DNA polymerase master mix (Canvax, Spain), consisting of 2 × PCR buffer, 0.4 mM of each dNTP, 4 mM MgCl_2_, and 0.1 U Taq DNA polymerase/μL. The PCR conditions followed those reported in [12]. After their visualization by gel electrophoresis, the PCR products were purified using ExoSAP-IT (USB Corp., OH, USA), sequenced using the Big Dye Terminator v3.1 reaction cycle sequencing kit (Applied Biosystems, Foster City, CA, USA), and migrated in an AB 3130 sequencer (Applied Biosystems) at the CACTI sequencing facilities (Universidade de Vigo, Spain). The obtained partial gene sequences from the amplified LSUrRNA were deposited in GenBank. The accession numbers are provided in Appendix A in the online version of this article.

### 4.4. Morphological Study of Gambierdiscus 

#### 4.4.1. Morphometric Analysis

Cell morphology determinations were based on measurements of two thecal plates: the second apical (2′) plate, located on the epitheca, and the second antapical (2″″) plate, on the hypotheca (Figure 7). For the 2′ plate, the lengths of the 2′/4′ (M1 in Figure 7a), 2′/1″ (M2 in Figure 7a), 2′/3″ (M3 in Figure 7a), and 2′/3′ (M4 in Figure 7a) sutures were measured. For the 2″″ plate, the maximum length of the plate (L in Figure 7b) and the width measured from the middle of the left side to the middle of the right side of the plate (W in Figure 7b) were determined. Based on those measurements, the following three parameters were evaluated: (1) R1, defined as the ratio of M2/M3, was used in an assessment of the rectangular vs. the hatchet shape of the 2′ plate; (2) R2, defined as the ratio of M4/M1, represented the position of Po in the lateral edge of the 2′ plate and, therefore, the degree of eccentricity of Po in the cell; and (3) R3, defined as the ratio of L/W, as an indicator of the elongation of the 2″″ plate. Cell depth (D), corresponding to the dorso-ventral diameter, and cell width (W), corresponding to the transverse diameter, were also measured. All measurements were made on calcofluor-stained cells using digital imaging software (ZEN lite, ZEISS Microscopy) and an epifluorescence microscope (Leica DMLA, Wetzlar, Germany) equipped with a UV light source and an AxioCam HRc (Carl Zeiss, Jena, Germany) digital camera. 

The thecal tabulation system [43] used in the present study was based on a modification of the Kofoid system as described by [44]. Thus, in our study, the first precingular plate (1′) was considered to be a small plate that does not contact Po (Figure 7a). This plate is regarded as the 1″ plate in other *Gambierdiscus* studies which consider, on other hand, the 4′ plate as the 1′ plate. The nomenclature used in our study has also been used in studies of others species, such as *Gonyaulacales*, and allows comparisons with species and genera sharing the same plate formula: Po, 4′, 6″, 5‴, and 2″″. Following the same criteria, although the sulcal posterior plate (Sp) in *Gambierdiscus, Coolia*, and *Ostreopsis* is outside the sulcus, it must be considered as Sp, such that the 1p plate described in previous reports of *Gambierdiscus* is herein denoted as the 2″″ plate (Figure 7b) [17,44].

#### 4.4.2. Cultured Cells

The morphological study was carried out on cultured cells of *Gambierdiscus* species identified by sequencing of the D8–D10 regions of the LSUrRNA gene, as described in Section 4.3. All of the strains used in this study were obtained from the isolation of cells in samples collected from the Canary Islands. Measurements were performed on at least 62 cells of each species (1–3 strains according to the availability of strains of each species) (Appendix A). 

#### 4.4.3. Field Samples

Morphological analyses were also performed on individual cells isolated from epiphytic samples taken from La Gomera (see Section 4.1). Measurements of the epitheca and hypotheca of the same specimen were made by placing individual cells between two coverslips, which allowed them to be observed and photographed from their apical and antapical views. The morphologies of a total of 42 cells obtained from seven samples of three stations, Santiago beach (4), Charco Condesa (32), and La Cueva beach (4) (Table 1), were determined. 

### 4.5. Comparison of the Morphological versus Genetic Data 

*Gambierdiscus* species identified morphologically and genetically were compared with respect to their abundances. Morphology results were obtained as described in Section 4.4.3 for 42 cells of *Gambierdiscus* and genetic data from 33 cultures of *Gambierdiscus* isolated from the same samples used in the morphological study: Santiago beach (8 isolates), Charco Condesa (14 isolates), and La Cueva beach (11 isolates). These cultures corresponded in part to those used to evaluate *Gambierdiscus* off La Gomera island (Section 4.1).

### 4.6. Statistical Analysis

A principal component analysis (PCA) was performed to analyze the data describing the composition of epiphytic dinoflagellates. It was conducted using logarithmically transformed cell concentrations and the statistical software package SPSS. The Kaiser–Meyer–Olkin measure of sampling adequacy was 0.69, and Bartlett’s test of sphericity, which tests for the presence of correlations among variables, was significant at *p* < 0.001.

Statistical analyses comparing the morphological parameters of the different *Gambierdiscus* species were performed using the statistical software package SPSS. A Shapiro–Wilks test (*p* > 0.05) showed that the variables were normally distributed. However, in a Levene’s test there was no homogeneity of variance among the variables, such that a non-parametric rank-based test (Kruskal–Wallis) was used in the species comparisons.

## Figures and Tables

**Figure 1 toxins-11-00423-f001:**
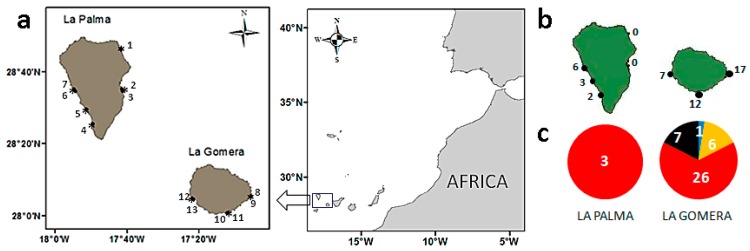
(**a**) Sampling localities in the coasts of La Palma and La Gomera (Canary Islands). Station numbers and their identification are provided in Table 1. (**b**) Average abundance of *Gambierdiscus* spp. (cells·g^−1^ macrophyte wet weight) in La Palma and La Gomera. (**c**) Number of sequences retrieved from individual *Gambierdiscus* species in molecular analyses of samples from each island. *G. australes* (blue), *G. caribaeus* (yellow), *G. silvae* (black), and *G. excentricus* (red).

**Figure 2 toxins-11-00423-f002:**
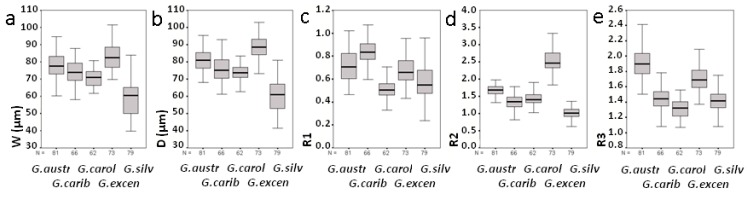
Box plots of cell width (**a**), cell depth (**b**), and parameters R1 (**c**), R2 (**d**), and R3 (**e**) used in the morphological analyses (see Section 4.4.1).

**Figure 3 toxins-11-00423-f003:**
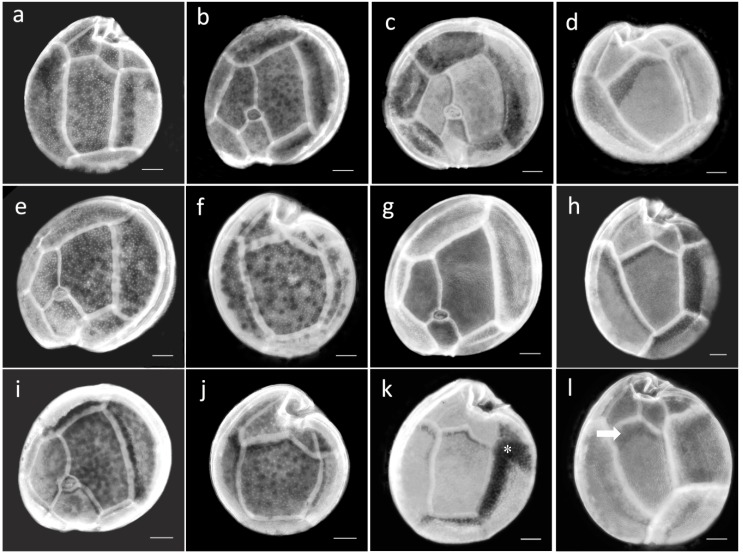
Photographs of cultured cells of the five species of *Gambierdiscus* reported from the Canary Islands and including their thecal plates. Hypotheca (**a**) and epitheca (**b**) of *G. australes*. Epitheca (**c**) and hypotheca (**d**) of *G. caribaeus.* Epitheca (**e**) and hypotheca (**f**) of *G. carolinianus*. Epitheca (**g**) and hypotheca (**h**) of *G. excentricus*. Epitheca (**i**) and hypotheca (**j**) of *G. silvae.* For the hypothecae of *G. australes* (**k**), the difference in their staining intensity (asterisk) reveals that they derived from a recently divided cell. The hypotheca of *G. excentricus* (**l**) has a less dense upper end of the 2″″ plate (arrow). Scale bars = 10 µm.

**Figure 4 toxins-11-00423-f004:**
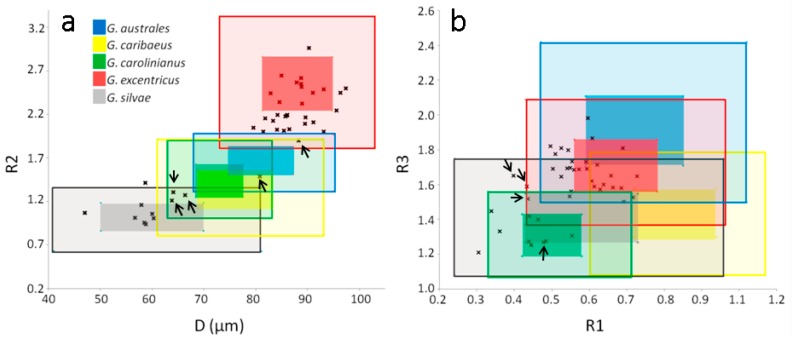
Scatterplots showing the relationships between D (cell depth) and R2 values (**a**) and between R1 and R3 values (**b**) in field cells of *Gambierdiscus* (black dots). The colored squares represent the ranges for standard deviation limits (dark colors) and maximum–minimum (light colors) of D vs. R2 (**a**) and R1 vs. R3 (**b**) as determined in the morphological study of cultured cells. The arrows indicate cells whose values are particularly mentioned in the text (Section 2.4).

**Figure 5 toxins-11-00423-f005:**
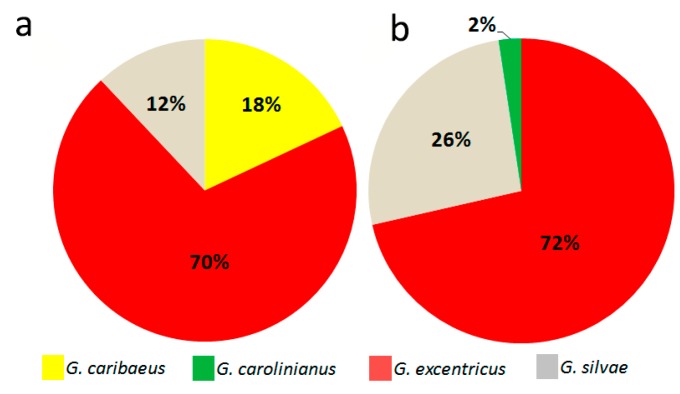
Abundance (%) of the *Gambierdiscus* species identified in a genetic analysis (**a**) and in a morphological study (**b**) from the La Gomera island samples.

**Figure 6 toxins-11-00423-f006:**
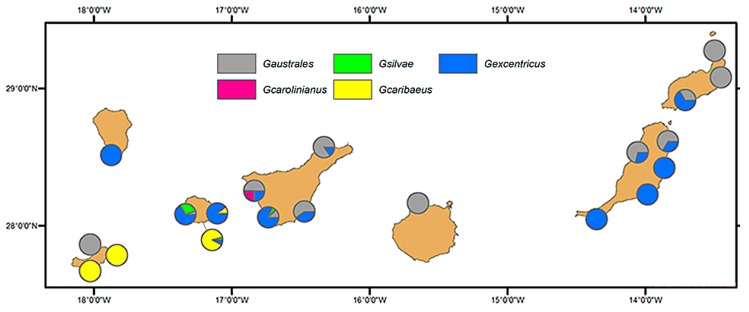
Geographical distribution of the different species of *Gambierdiscus* in the Canary Islands.

**Figure 7 toxins-11-00423-f007:**
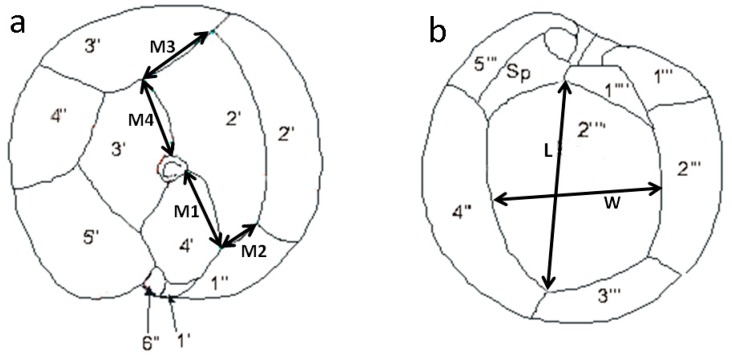
Drawings of an epitheca (**a**) and hypotheca (**b**) of *Gambierdiscus*, showing the measurements used in the morphological study.

**Table 1 toxins-11-00423-t001:** Location of sampling stations in La Palma and La Gomera Islands.

Station	Island	Locality	Coordinates	Type of Sample
1	La Palma	Charco azul	28.81089, −17.7642	macrophyte
2	La Palma	Cancajos beach	28.65232, −17.75951	macrophyte
3	La Palma	Caleta Ballena	28.64465, −17.7573	macrophyte
4	La Palma	La Zamora beach	28.51485, −17.87486	macrophyte
5	La Palma	Charco verde	28.57261, −17.90008	macrophyte
6	La Palma	Tazacorte harbor	28.6437, −17.94325	net
7	La Palma	Tazacorte beach	28.65157, −17.94887	macrophyte
8	La Gomera	La Cueva beach	28.09081, −17.10565	macrophyte
9	La Gomera	San Sebastián harbor	28.089, −17.10767	net
10	La Gomera	Santiago beach	28.02732, −17.1979	macrophyte
11	La Gomera	Santiago harbor	28.02629, −17.19755	net
12	La Gomera	Valle Rey harbor	28.08069, −17.33259	net
13	La Gomera	Charco Condesa	28.08397, −17.33653	macrophyte

**Table 2 toxins-11-00423-t002:** Rotated component matrix from the principal component analysis (PCA) for epiphytic dinoflagellates in the islands of La Gomera and La Palma.

	Component	
	1	2
*Prorocentrum*	0.568	-
*Coolia*	0.583	-
*Sinophysis*	0.709	-
*Ostreopsis*	-	0.685
*Vulcanodinium*	0.524	-
*Heterocapsa*	-	0.792
*Scripsiella*	-	0.625
*Gambierdiscus*	0.706	-

**Table 3 toxins-11-00423-t003:** Mean, standard deviation, and maximum and minimum values of cell size measurements (µm) and morphological parameters (see material and methods) of *Gambierdiscus* species.

Parameter	Species	Mean	Std	Max	Min
D (cell depth)	*G. australes*	81	6.3	95	68
	*G. caribaeus*	76	7.2	93	61
	*G. carolinianus*	73	4.6	83	63
	*G. excentricus*	88	6.8	103	73
	*G. silvae*	60	9.9	81	41
W (cell width)	*G. australes*	78	7.5	95	60
	*G. caribaeus*	74	6.7	88	58
	*G. carolinianus*	71	4.9	81	62
	*G. excentricus*	83	7.9	101	70
	*G. silvae*	59	9.7	84	40
R1	*G. australes*	0.72	0.13	1.02	0.47
	*G. caribaeus*	0.83	0.11	1.07	0.60
	*G. carolinianus*	0.50	0.08	0.71	0.33
	*G. excentricus*	0.67	0.11	0.96	0.43
	*G. silvae*	0.58	0.15	0.96	0.24
R2	*G. australes*	1.67	0.16	1.97	1.32
	*G. caribaeus*	1.34	0.22	1.92	0.82
	*G. carolinianus*	1.44	0.19	1.90	1.02
	*G. excentricus*	2.55	0.31	3.33	1.82
	*G. silvae*	1.02	0.16	1.36	0.63
R3	*G. australes*	1.91	0.20	2.42	1.50
	*G. caribaeus*	1.44	0.14	1.78	1.08
	*G. carolinianus*	1.31	0.12	1.56	1.07
	*G. excentricus*	1.71	0.15	2.09	1.37
	*G. silvae*	1.41	0.14	1.75	1.08

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
