# Peer review of "Ciguatera-Causing Dinoflagellate Gambierdiscus spp. (Dinophyceae) in a Subtropical Region of North Atlantic Ocean (Canary Islands): Morphological Characterization and Biogeography"

_toxins, 2019, doi:10.3390/toxins11070423_

Round 1

Reviewer 1 Report

This manuscript is a well written account of a well executed and much-needed study. Some minor revision is needed to make it more complete. The authors need to:

1) More completely describe the context of the situation in the Canary Islands regards ciguatoxins, update citation and references on relevant  papers published recently addressing ciguatoxins in the Canary Islands:

Estevez, Pablo, David Castro, Ana Pequeño-Valtierra, José M. Leao, Oscar Vilariño, Jorge Diogène, and Ana Gago-Martínez. "An attempt to characterize the ciguatoxin profile in seriola fasciata causing ciguatera fish poisoning in macaronesia." Toxins 11, no. 4 (2019): 221.

The following is also relevant and includes mention of a new hydroxyl metabolite of C-CTX1:

Estevez, Pablo, David Castro, J. Manuel Leao, Takeshi Yasumoto, Robert Dickey, and Ana Gago-Martinez. "Implementation of Liquid Chromatography tandem Mass Spectrometry for the Analysis of ciguatera fish poisoning in contaminated fish samples from Atlantic coasts." Food chemistry 280 (2019): 8-14.

2) Discuss more on the relevance of the study regards ciguatoxin production, particularly those that impact humans, for example:

Mention more about the fact that generally, CTX associated  algae like Gambierdiscus do not directly produce the potent toxins implicated in ciguatera. First CTXs are produced as different (less toxic) forms and are then metabolized in fish to the very toxic forms that cuase ciguatera. The authors need to make this clear.

First in the abstract:

"Dinoflagellates belonging to the genus Gambierdiscus produce ciguatoxins (CTXs), which accumulate in fish and subsequently cause ciguatera fish poisoning (CFP) in humans"

The above text is somewhat misleading becuase readers may believe that the algae produce the CFP-causing toxins directly.

"Dinoflagellates belonging to the genus Gambierdiscus produce ciguatoxins (CTXs), which are metabolized in fish  to more toxic forms subsequently causing ciguatera fish poisoning (CFP) in humans"

And in the introduction:

"Gambierdiscus is a genus of benthic dinoflagellates that produces ciguatoxins (CTXs) and maitotoxins (MTXs). The transfer of these toxins up the food chain results in their accumulation in the tissues of fish and ultimately in ciguatera fish poisoning (CFP) in humans"

Replace with text similar to the following:

"Gambierdiscus is a genus of benthic dinoflagellates that produces ciguatoxins (CTXs) and maitotoxins (MTXs). The transfer of CTXs up the food chain results in their accumulation in the tissues of fish along with their metabolism to more toxic forms causing ciguatera fish poisoning (CFP) in humans"

Author Response

Thanks for the comments. All the reviewer's suggestions have been followed.

This manuscript is a well written account of a well executed and much-needed study. Some minor revision is needed to make it more complete. The authors need to:
1) More completely describe the context of the situation in the Canary Islands regards ciguatoxins, update citation and references on relevant papers published recently addressing ciguatoxins in the Canary Islands:
Estevez, Pablo, David Castro, Ana Pequeño-Valtierra, José M. Leao, Oscar Vilariño, Jorge Diogène, and Ana Gago-Martínez. "An attempt to characterize the ciguatoxin profile in seriola fasciata causing ciguatera fish poisoning in macaronesia." Toxins 11, no. 4 (2019): 221.
The following is also relevant and includes mention of a new hydroxyl metabolite of C-CTX1:
Estevez, Pablo, David Castro, J. Manuel Leao, Takeshi Yasumoto, Robert Dickey, and Ana Gago-Martinez. "Implementation of Liquid Chromatography tandem Mass Spectrometry for the Analysis of ciguatera fish poisoning in contaminated fish samples from Atlantic coasts." Food chemistry 280 (2019): 8-14.
Answer:
Done. We have added few paragraphs in the introduction and that references (line 71) as well as another relevant manuscript on G. polynesiensis as the editor indicated .
2) Discuss more on the relevance of the study regards ciguatoxin production, particularly those that impact humans, for example:

Mention more about the fact that generally, CTX associated algae like Gambierdiscus do not directly produce the potent toxins implicated in ciguatera. First CTXs are produced as different (less toxic) forms and are then metabolized in fish to the very toxic forms that cuase ciguatera. The authors need to make this clear.
Answer:
Done. We think that now the first paragraph from introduction is clearer. In relation with the comment about discussing more on the relevance regarding ciguatoxin production. We have added one paragraph in introduction saying that there are no description of the CTX profile of Gambierdiscus with the exception of G. polynesiensis (lines 39-41), and other on the ciguatoxins from fishes from the studied area (lines 69-71). We think that is enough since the object of the work are not the toxins but the morphology and the biogeography. We would like to add that although the focus of the work is outside the scope of the journal, it is within the special issue "Benthic ecosystems": Taxonomy, genetic diversity, ecology and toxicity of biofilms dominated by potentially-toxic cyanobacteria and microalgae and on risk assessment and management associated with such assemblage.

First in the abstract:
"Dinoflagellates belonging to the genus Gambierdiscus produce ciguatoxins (CTXs), which accumulate in fish and subsequently cause ciguatera fish poisoning (CFP) in humans"
The above text is somewhat misleading becuase readers may believe that the algae produce the CFP-causing toxins directly.
"Dinoflagellates belonging to the genus Gambierdiscus produce ciguatoxins (CTXs), which are metabolized in fish to more toxic forms subsequently causing ciguatera fish poisoning (CFP) in humans"
And in the introduction:
"Gambierdiscus is a genus of benthic dinoflagellates that produces ciguatoxins (CTXs) and maitotoxins (MTXs). The transfer of these toxins up the food chain results in their accumulation in the tissues of fish and ultimately in ciguatera fish poisoning (CFP) in humans"
Replace with text similar to the following:
"Gambierdiscus is a genus of benthic dinoflagellates that produces ciguatoxins (CTXs) and maitotoxins (MTXs). The transfer of CTXs up the food chain results in their accumulation in the tissues of fish along with their metabolism to more toxic forms causing ciguatera fish poisoning (CFP) in humans"
Answer:
We have made the changes suggested both in the abstract and introduction.

Reviewer 2 Report

The manuscript, ‘Ciguatera-causing dinoflagellate Gambierdiscus spp. (Dinophyceae) in a subtropical region of North Atlantic Ocean (Canary Islands): morphological characterization and biogeography’ is a well-written study describing the abundance and distribution of previously described Gambierdiscus spp. from the Canary Islands. The authors have analyzed the ecology and morphology of Gambierdiscus spp. quite thoroughly. However, there are some major issues with the manuscript.

The study claims the identification of 5 Gambierdiscus species from the Canary Islands but has not shown any phylogenetic evidence. Such studies would highly benefit from a phylogenetic tree to clearly visualize the phylogenetic diversity. Please consider.

The manuscript doesn’t do justice to recent studies done on Gambierdiscus spp. since they have not been cited. I can understand this is not a review but please use some more recent references to encompass the magnitude of Gambierdiscus/ciguatera studies occurring around the globe. Also, several species identified in this study are non-CTX producers and should be discussed in more detail.

 The quality of scientific writing is good for the majority of the manuscript; however, several sections require thorough proof-reading/re-writes.

Table 3 and Figure 2 are showing the same thing. Table 3 can be moved to the supplementary section.

The manuscript has a very strong ecological and morphological focus but has not analyzed the toxin profile of any of the cultured isolates. While most of others issues, I have mentioned are minor, I believe that a journal such as ‘Toxins’ should not be targeted if there is no toxin profile data in the manuscript. I would sincerely suggest the authors add LC-MS/toxicity data of some cultured isolates into the manuscript and attempt to re-submit or try sending the manuscript to other journals such as J Phycol., Harmful Algae, TEJP, PLoS or Marine pollution bulletin.

Author Response

Thanks for the comments. In relation to the comment of this reviewer about our work is not within Toxin scope, we would like to say that we have chosen Toxins journal because this is a special issue dedicated to benthic ecosystems. And that as it is explained on the journal's website covers more than the toxins.

Reviewer 2
The manuscript, ‘Ciguatera-causing dinoflagellate Gambierdiscus spp. (Dinophyceae) in a subtropical region of North Atlantic Ocean (Canary Islands): morphological characterization and biogeography’ is a well-written study describing the abundance and distribution of previously described Gambierdiscus spp. from the Canary Islands. The authors have analyzed the ecology and morphology of Gambierdiscus spp. quite thoroughly. However, there are some major issues with the manuscript.
The study claims the identification of 5 Gambierdiscus species from the Canary Islands but has not shown any phylogenetic evidence. Such studies would highly benefit from a phylogenetic tree to clearly visualize the phylogenetic diversity. Please consider.
Answer:
As the editor said this is not a description of new species. We consider phylogenetic tree is not needed.

The manuscript doesn’t do justice to recent studies done on Gambierdiscus spp. since they have not been cited. I can understand this is not a review but please use some more recent references to encompass the magnitude of Gambierdiscus/ciguatera studies occurring around the globe. Also, several species identified in this study are non-CTX producers and should be discussed in more detail.
Answer:
We do not agree with this comment. We think that the references are fully adapted to the context of the manuscript which is the five species of Gambierdiscus from Canary Islands. Of course there are more bibliography since there are 16 species described in total. But that comes out of the context of our work.

The quality of scientific writing is good for the majority of the manuscript; however, several sections require thorough proof-reading/re-writes.
Table 3 and Figure 2 are showing the same thing. Table 3 can be moved to the supplementary section.
Answer:
We think that information is relevant because in the box-plot figure the limits are not visible.

The manuscript has a very strong ecological and morphological focus but has not analyzed the toxin profile of any of the cultured isolates. While most of others issues, I have mentioned minor, I believe that a journal such as ‘Toxins’ should not be targeted if there is no toxin profile data in the manuscript. I would sincerely suggest the authors add LC-MS/toxicity data of some cultured isolates into the manuscript and attempt to re-submit or try sending the manuscript to other journals such as J Phycol., Harmful Algae, TEJP, PLoS or Marine pollution bulletin.
Answer:
It is true that this work does not correspond to the scope of the journal Toxins but we think is within the scope defined for the special issue "Benthic ecosystems", as it is explained in the website of the journal: All papers dealing with the taxonomy, genetic diversity, ecology and toxicity of biofilms dominated by potentially-toxic cyanobacteria and microalgae and on risk assessment and management associated with such assemblages will be considered in this Special Issue.

Reviewer 3 Report

Authors reports morphological characterization and biogeography of Gambierdiscus spp. collected at Canary Islands. The results are suitable for publication inToxins after minor revisions. 1. Readers of Toxins are interesetd in toxicity of Gambierdiscus spp. Are there any results about toxicity ? 2. Page 5, Table 3. Use italic for species.

Author Response

Thanks for the comments. In relation to the comment of this reviewer about our work is not within Toxin scope, we would like to say that we have chosen Toxins journal because this is a special issue dedicated to benthic ecosystems. And that as it is explained on the journal's website covers more than the toxins.

Reviewer 3 Readers of Toxins are interesetd in toxicity of Gambierdiscus spp. Are there any results about toxicity ?

Answer: We have added some information on toxins of Gambierdiscus in the introduction as other reviewer and the editor suggested. However, I would like to explain that we think is within the scope defined for the special issue "Benthic ecosystems", as it is explained in the website of the journal: All papers dealing with the taxonomy, genetic diversity, ecology and toxicity of biofilms dominated by potentially-toxic cyanobacteria and microalgae and on risk assessment and management associated with such assemblages will be considered in this Special Issue.

2. Page 5, Table 3. Use italic for species.

Answer: Done